# Quantile granger causality between clean energy and tourism stock indices: Evidence from regional markets

Ozge Demirkale[1]*, Naime Irem Duran[2]

1 Istanbul Aydin University, Faculty of Economics and Administrative Sciences, Department of Business Administration, Istanbul, Turkey, 2 Beykent University, Faculty of Economics and Administrative Sciences, Capital Markets Department, Istanbul, Turkey

* ozgedemirkale@aydin.edu.tr

## Abstract

As two pillars of the green transition, clean energy and tourism have gained growing strategic prominence in the landscape of sustainable finance, warranting a deeper investigation into their financial interdependencies. However, empirical research exploring their interaction in financial markets, particularly from a regional perspective, remains relatively limited. This study contributes to that objective by examining the predictive relationships between the WILDERHILL Clean Energy Index and tourism indices from the United States, Europe, China, and Australia. Using monthly data from 2010 to 2023, the analysis applies quantile Granger causality and wavelet coherence methods to capture asymmetric and time-varying dynamics. Additionally, a structural VAR model is used to assess region-specific responses to clean energy shocks. While conventional Granger tests do not indicate significant linkages, quantile-based approaches uncover heterogeneous connections that emerge under extreme market conditions. The findings reveal increasing co-movement between clean energy and tourism sectors and emphasize the importance of distribution-sensitive and regionally contextualized approaches in guiding investment and policy-making strategies.

## 1. Introduction

Climate change, environmental degradation, and sustainable development goals are key issues in global economic policies today. In this respect, transitions towards clean energy sources are one of the most striking areas that shape the course of environmental and financial systems. In particular, the transformation of energy markets has been accelerated in light of global climate change and SDGs, which has resulted in an increased trend towards clean energy. Not only does the energy sector heavily depend on the reduction in the use of fossil fuels and a wide dispersion of renewable energy technologies, but sectors like financial markets and tourism are also seriously affected.

**Data availability statement:** All relevant data are within the manuscript and its Supporting Information files.

**Funding:** The author(s) received no specific funding for this work.

**Competing interests:** The authors have declared that no competing interests exist.

Tourism is essential for economic growth, employment, and foreign exchange earnings, yet it remains highly vulnerable to climate change impacts such as rising sea levels, desertification, and drought [1]. Adopting cleaner energy within tourism operations helps reduce the sector's environmental footprint and enhances its resilience to climate risks [2].

Clean energy investments including solar, wind, hydroelectric, and geothermal have rapidly grown, reaching 1.7 trillion USD globally in 2023 (IEA). This increase is supported by policy initiatives such as the European Green Deal, the Paris Agreement [3,4], and sustainable finance mechanisms like green bonds and sustainable indices provided by stock exchanges [5].

The COVID-19 pandemic experienced unprecedented fluctuations in tourism and renewable sectors. With borders reopening and a growing demand for ecotourism in post-2022, a slow and steady restoration in tourism started taking shape. According to estimates by the World Tourism Organization (UNWTO), tourism in 2023 will rise to 88% of its 2019 level and overtake the 2019 level in 2024.

Amid increasing uncertainty in the marketplace, expansion, and stability in the clean energy sector have continued to encounter substantial impediments [6]. Notwithstanding, financing for developing clean energy comes from a principal source, namely, the Clean Energy Index [7]. In relation, current studies have a range of studies investigating the nexus between financial markets and clean energy indices [8–15]. In consonance with such studies, a general observation is that, even with a positive contribution towards clean energy stocks, the pandemic took a toll both in financial and energy markets.

This study aims to empirically investigate interrelationships between European, US, Australian, and Chinese tourism stock market indices and the WilderHill Clean Energy Index through econometric models. In this case, interrelationships between tourism and energy sectors, investments in clean energy and tourism, and their financial and economic performance impact are analyzed. Utilizing tourism stock market indices in Europe, the US, Australia, and China is an important move towards analyzing specific tourism and energy sectors' dynamics in these countries. In addition, not only do considered indices in this work explore geographical diversity in interrelationships between tourism and energy sectors, but through them, one can evaluate the financial performance impacts of these sectors in an international comparative environment.

Europe is exceptional because it is the global front for sustainability in tourism and energy policies. As a result, the European Union continues to raise its investment in renewable energy with its Green Deal to achieve carbon neutrality by 2050, which also calls upon the tourism industry to be friendly. It supports renewable energy use, such as in Germany, Spain, and France, and initiatives such as sustainable tourism. It has been noted that the financial performance related to environmental sustainability and the tourism industry is firmly connected in Europe. Australia is also known for its eco-tourism potential and renewable energy projects [1].

It is worth noticing that the United States, ranked among the leading countries in the tourism sector, is distinguished by the scale of investments and pioneering

energy solutions. Significant investments by firms like Disney and Hilton develop closer ties between the tourism sector and the energy market. According to findings, it is observed that stocks and ESG ratings have been positively influenced by tourism companies investing in renewable energy projects. This confirms the strong financial interaction between the tourism and energy markets [16–19]. Other areas of interest in our research include China's rapidly developing tourism industry and increasing energy use. Conserving cultural heritage and ecological initiatives also provides an environment enabling the tourism industry. A recent study examined 30 provinces in China using data from 2007 to 2021 and concluded that, generally, tourism development negatively affected high-quality economic growth. However, the result was positive for more developed regions [20].

Despite increasing attention to sustainable finance, few studies have empirically explored direct financial linkages between clean energy and tourism sectors. This study extends the existing literature by focusing on the relationship between clean energy and tourism sectors, particularly in major world economies. Earlier studies have established various linkages between tourism stocks, economic factors, and financial markets. The novelty of this study lies in its focus on a direct analysis of the predictive association between the clean energy index and the tourism stock performance in the United States, China, Australia, and the European Union.

While previous studies have independently examined clean energy and tourism stock market dynamics, empirical analyses exploring their financial interconnections remain limited. Given recent global developments such as COP-26 commitments, growing ESG investment trends, and post-COVID-19 economic recovery efforts, understanding the financial interactions between these two sectors is crucial for policy and investment decisions.

Accordingly, the study addresses the following research questions: (1) Is there a long-run and short-run predictive relationship between the WilderHill Clean Energy Index and tourism stock indices in the United States, Europe, China, and Australia? (2) Do this relationship's direction and causality dynamics vary by region from unidirectional to bidirectional? (3) What regional differences might determine the character of the relationship between clean energy and tourism indices? (4) How have global economic and non-economic shocks, for instance, COVID-19, transformed the relationship between the two sectors?

Based on these research aims, the following hypotheses are formulated

$H_1$: Clean energy stock returns Granger-cause tourism stock returns, particularly under extreme market conditions (i.e., in upper and lower quantiles).

$H_2$: The direction and strength of Granger causality between clean energy and tourism stock returns differ across regions (Europe, US, China, Australia).

$H_3$: In regions with stronger sustainable tourism practices (e.g., Europe and Australia), the positive effect of clean energy on tourism stocks is more pronounced.

$H_4$: Global shocks such as COVID-19 amplify the asymmetric relationship between clean energy and tourism sectors over time.

This article is organized as follows: Section 2 reviews the relevant literature, while Section 3 introduces the methods and dataset used. Section 4 presents the empirical results. Finally, Section 5 summarizes the findings, outlines policy implications, and discusses the study's limitations, providing insights for future research.

## 2. Background literature

### 2.1 Tourism investments, economic growth, and renewable energy

In the dynamic structure of the world economy, tourism investments represent one of the most important leading economic growth factors. At the same time, their interaction with renewable energy sources within the goals of sustainable

development has also taken an increasingly central place in the attention of researchers. In this regard, the interaction between the potential of tourism sector investments to stimulate economic growth and environmental sustainability has emerged as an important area of research in the literature [20–26].

Isik et al. explored the nexus between tourism development, renewable energy consumption, and economic growth in the United States, France, Spain, China, Italy, Turkey, and Germany within a bootstrap panel Granger causality model framework throughout 1995–2012. Their findings show that the relationship is highly country-specific: tourism development and economic growth are interdependent in Germany, while tourism drives growth in China and Turkey, but not in Spain, where the reverse effect is observed [21]. Similarly, Lu et al. emphasized that renewable energy and tourism investments should be considered joint engines of sustainable tourism development, particularly across G20 economies [22]. Ben Jebli et al. focusing on 22 Central and South American countries (1995–2010), found that tourism, renewable energy, and FDI jointly contribute to emissions reduction, reinforcing the synergy between environmental and economic goals [23]. Avcı et al. investigating the same period (1995–2019) in the top 15 most visited countries, reached more cautious conclusions. They showed that, despite green technological innovation and tourism growth, financial development and economic expansion tend to increase $CO_2$ emissions, pointing to a deterioration in environmental quality. This suggests that the scale effect of growth may offset technological gains in emissions mitigation [26].

Other recent studies focus on emerging economies. Gyamfi et al. found that energy investment, economic growth, and eco-friendly tourism help reduce $CO_2$ emissions in E7 countries during 2000–2018 [24], whereas Li et al. documented that tourism has adverse effects on high-quality economic development (HQED) in China, except in more advanced regions [20]. These findings indicate that tourism's environmental performance may vary with regional development levels and institutional quality. Sreenu emphasized the role of green finance, showing that green bonds support environmental sustainability in India over the long run, while inflation (CPI) undermines it in both the short and long term [25].

Taken together, these studies reflect mixed but complementary perspectives: while tourism and renewable energy often align with sustainability, their impact is mediated by factors such as financial maturity, regional development, and policy instruments. This diversity highlights the need for regionally nuanced empirical models that account for asymmetric and long-run dynamics. Moreover, most studies employ static or panel-mean estimation frameworks, with limited attention to how relationships vary across different market states or quantile levels a methodological gap this study directly addresses.

## 2.2. The relationship between clean energy and stock indices

Amid the accelerating transition to a low-carbon economy, the relationship between clean energy investments and financial markets particularly through the lens of stock index performance has become a prominent theme in sustainable finance literature. Understanding these dynamics is essential for designing portfolios that integrate environmental criteria while navigating market volatility [8–15,27].

Ferreira and Loures found that clean energy indices exhibit stronger time-series persistence and lower oil price sensitivity compared to traditional market indices such as the NYSE [9]. In contrast, Wang et al. and Wan et al. emphasized the pandemic-induced shift in investor preferences, observing that capital moved toward solar and renewable energy stocks as a diversification strategy [10,11]. Wan et al. reported a positive COVID-19 effect on clean energy stocks but negative impacts on broader energy markets, whereas Hemrit and Benlagha confirmed this resilience by showing a significant positive effect of the pandemic on renewable energy indices [11,12]. While both studies acknowledge pandemic resilience, the latter provides stronger causal evidence, whereas the former relies primarily on return-based correlations, underscoring the need for more distribution-sensitive causal frameworks. Ghabri et al. added to this evidence by identifying a negative correlation between fossil fuel prices and clean energy prices, supporting the substitution effect [14]. Shahbaz et al. employing various causality tests using high-frequency data from 2005 to 2021, showed that the responsiveness of clean energy stocks to oil and market shocks is asymmetric and state-dependent [13]. Similarly, Alam et al. found that R&D investments and developed stock markets in OECD countries positively affect clean energy consumption and contribute to environmental sustainability [8].

From a risk management perspective, Ghosh revealed that although pandemic-related uncertainty had a negative effect on clean energy indices, these assets still function effectively as diversification tools in volatile conditions [15]. Most recently, Athari and Kirikkaleli examined the dynamic relationship between Climate Policy Uncertainty (CPU) and the Renewable Energy and Clean Technology Index (RECT) in Canada [27]. Their findings demonstrate that the direction and strength of this relationship vary over time, with CPU exerting significant influence on clean energy stock prices especially in short- and medium-term periods. These findings reinforce the notion that clean energy stocks are not immune to climate-related uncertainty, and that these effects may be masked in linear or full-sample average models an issue our quantile-based approach seeks to overcome. This underscores the broader vulnerability of green assets to climate-related policy signals.

Taken together, these studies converge on a common finding: while clean energy equities offer promising diversification and sustainability advantages, their performance is highly context-dependent. Investor sentiment, policy uncertainty, and global shocks such as pandemics or geopolitical tensions mediate these effects, suggesting that linear models may fail to fully capture the complex, time-varying nature of this financial-environmental interface.

### 2.3  The relationship between clean energy and tourism stock indices

Clean energy and tourism are among the fastest-growing sectors aligned with global sustainability objectives. However, despite their parallel growth, the financial linkages between these sectors—particularly at the equity index level have received limited scholarly attention. Understanding how clean energy developments influence tourism stock markets may offer novel insights into sustainable finance, sectoral co-movements, and policy-driven investment behavior [28–30].

Calderón-Vargas et al. focusing on Peru, emphasized that despite institutional gaps in sustainable tourism policy, the country's solar and wind energy potential supports initiatives such as sustainable museums. This study highlights the role of renewable energy infrastructure in facilitating tourism development, albeit through a qualitative, project-level lens [28].

More directly addressing financial markets, Zeng et al. analyzed the correlation between seven clean energy indices and the Dow Jones U.S. Travel and Leisure Index using daily data from 2014 to 2023. Their results showed a consistently high correlation, suggesting that clean energy market fluctuations may have meaningful predictive power over tourism sector performance in the U.S [29]. However, Zeng et al.'s correlation-based approach does not fully uncover the directional or asymmetric nature of the linkages an issue addressed more explicitly by Wang et al. using causality tests, albeit still limited to mean-based estimates. Expanding the regional scope, Wang et al. examined the causal relationship between clean energy indices and tourism stock indices across the United States, Europe, and China from January 2018 to December 2022 [30]. Their econometric findings indicate a statistically significant influence of clean energy index movements on regional tourism equities, with notable variation across countries. In particular, the U.S. and Europe exhibited stronger directional effects than China, potentially reflecting differences in sustainability integration and investor responsiveness.

In summary, while the literature remains scarce, emerging studies point to a growing financial interdependence between clean energy and tourism sectors. However, most existing research focuses on either correlation or single-country analyses, often overlooking time-varying and quantile-specific dynamics. This underscores the need for broader, multi-regional approaches using advanced methods that capture asymmetry and non-linearity in sectoral interactions. As most of these studies are confined to single-country cases or limited data periods, there remains a need for high-frequency, multi-country analyses that incorporate both time and distributional Dynamics an analytical gap this study aims to fill.

### 2.4  Literature gap

Although the literature has extensively examined the individual impacts of tourism and clean energy on economic growth, emissions, and financial markets, studies exploring their direct interaction at the stock market level remain limited. Most

existing research either focuses on a single region or employs correlation-based methods without fully accounting for time-varying dynamics, asymmetric effects, or market-specific volatility.

In particular, empirical investigations linking clean energy indices to tourism equity performance across multiple regions are notably scarce. There is a lack of comprehensive, high-frequency, and regionally comparative studies that assess how clean energy developments influence tourism stocks under different structural and economic conditions. Furthermore, the majority of studies overlook how global shocks such as the COVID-19 pandemic alter these inter-sectoral linkages over time.

This study addresses this gap by analyzing the dynamic, quantile-specific, and frequency-domain relationships between the WilderHill Clean Energy Index and tourism stock indices in the United States, Europe, China, and Australia, offering a broader and more nuanced understanding of how sustainability-driven markets interact across global financial systems. While existing research provides valuable insights into tourism energy linkages and their role in sustainable development, empirical models rarely integrate both time-scale and distributional asymmetries across multiple regions. This study responds to this methodological and contextual gap by employing quantile Granger causality and wavelet coherence methods on monthly data from four major economic regions. In doing so, it contributes a novel perspective to the literature by capturing context-dependent, asymmetric causal linkages between clean energy and tourism stock indices relationships that are otherwise obscured in conventional linear or static models.

From an economic perspective, the relationship between clean energy and tourism stock indices can be framed through multiple transmission channels. First, investor sentiment driven by ESG criteria may cause capital reallocation from environmentally vulnerable sectors like tourism to cleaner alternatives during periods of environmental or financial stress. Second, clean energy developments signal policy shifts and long-term sustainability trends that can influence tourism-related investment decisions, especially in regions with strong eco-tourism agendas. Third, increased clean energy adoption within tourism operations may reduce sectoral costs and risks associated with fossil fuel dependency, thereby enhancing the financial attractiveness of tourism firms. These channels provide a theoretical basis for the observed empirical linkages and help explain the regional and quantile-dependent heterogeneity found in our results. These transmission channels are conceptually aligned with the theoretical framework proposed by Pástor, Stambaugh, and Taylor who show that investors require compensation for holding ESG-tilted assets and that ESG characteristics influence asset prices through changes in investor preferences and expectations about future cash flows [31]. In this context, clean energy indices may act as forward-looking indicators reflecting sustainability-driven capital flows, which in turn can affect tourism sector valuations particularly in regions where ESG alignment is central to investment decisions.

## 3. Data and model specifications

The analysis period covers data from January 2010 to December 2023. Our dataset includes tourism sector indices from the stock markets of the United States (US), Europe (EURO), China (CN), and Australia, as well as the Wilder Clean Energy Index. The sample period extends from January 1, 2010, to December 31, 2023. The year 2010 was selected as the starting point as it represents the beginning of the post-global financial crisis recovery period and reflects structural shifts that affected both the tourism and clean energy sectors.

Although partial price data were available for 2024, this year was excluded from the analysis to ensure consistency across all indices in terms of data completeness and monthly frequency. As not all indices had fully updated and verified monthly data for 2024, the year 2023 was selected as the most recent complete and reliable observation period. The Wilder Clean Energy Index data were obtained from WilderShares [32]. The tourism-related indices (DJUSTT, STOXXR, FTXIN, and AXHRKD) were retrieved from Investing.com [33], and all series were collected at a monthly frequency. The analyses were conducted using R and E-Views software. All series were transformed into their logarithmic forms. Table 1 presents descriptive information about the variables used in the analysis.

**Table 1. Definitions and Codes.**

| Variable | Symbol |
|---|---|
| WILDERHILL Clean Energy Index | LWILDER |
| STOXX Europe 600 Travel & Leisure Index | LSTOXXR |
| Dow Jones U.S. Travel & Leisure Index | LDJUSTT |
| FTSE China A 600 Travel & Leisure | LFTXIN |
| S&P/ASX 300 Hotels Restaurants & Leisure | LAXHRKD |

The countries/regions selected for this study (the United States, Europe, China, and Australia) represent advanced economies with substantial influence over both global tourism flows and international financial markets. These regions serve as primary nodes of both domestic and international tourism demand. In addition, they were selected based on the availability of high-frequency, consistent, and reliable financial data. This selection strategy was adopted to maintain methodological rigor and enhance cross-regional comparability within the empirical analysis.

Descriptive statistics comprising the mean, standard deviation, skewness, and kurtosis for the time series utilized in this study are summarized in Table 2.

Table 2 presents the basic descriptive statistics for the index series used in the analysis. Among the average values, the highest mean is observed for the China tourism index (LFTXIN, 9.075), while the lowest corresponds to the clean energy index (LWILDER, 4.245). This reflects the overall differences in price levels across the indices during the study period. The standard deviation values, which indicate the degree of volatility, are relatively high for LWILDER (0.439) and LDJUSTT (0.409), suggesting that these series exhibit greater variability. In contrast, LSTOXXR (0.281) and LFTXIN (0.287) are comparatively more stable.

With respect to skewness, LWILDER and LFTXIN show positive (right-skewed) distributions, whereas LSTOXXR, LDJUSTT, and LAXHRKD exhibit negative (left-skewed) distributions. These patterns provide insights into the presence of extreme values and asymmetries in the data. Regarding kurtosis, all series display values below that of a normal distribution, indicating a relatively flatter distribution and lower probability of extreme values. Overall, Table 2 offers a summary of the statistical characteristics of the dataset, providing a general intuition about the structure and behavior of the index series included in the study.

The stationarity principle is highly pivotal in the analysis of time series since much of the econometric methods require time series data to be stationary to make statistical inferences accurate. A time series is stationary when its mean, variance, and autocovariance are constant over time. The presence of non-stationarity in a time series can lead to spurious regressions, rendering statistical inferences incorrect [34–36]. Therefore, before any econometric modeling or forecasting, it is essential to determine the stationarity properties of the data. In this study, the stationarity properties of the dataset were examined using the ADF, PP, and KPSS tests. A substantial stationarity test was conducted by comparing the results

**Table 2. Summary Statistics of the Series Used in the Analysis.**

| Statistic | LWILDER | LSTOXXR | LDJUSTT | LFTXIN | LAXHRKD |
|---|---|---|---|---|---|
| Mean | 4.245 | 5.275 | 6.254 | 9.075 | 8.824 |
| Median | 4.117 | 5.387 | 6.323 | 9.055 | 8.904 |
| Maximum | 5.541 | 5.628 | 7.010 | 9.709 | 9.400 |
| Minimum | 3.638 | 4.647 | 5.238 | 8.478 | 8.153 |
| Std. Dev. | 0.439 | 0.281 | 0.409 | 0.287 | 0.346 |
| Skewness | 0.842 | −0.766 | −0.714 | 0.044 | −0.523 |
| Kurtosis | 3.076 | 2.200 | 2.919 | 2.388 | 1.991 |

of these comparative tests and guided the resulting modeling decisions. The results of these tests played a significant role in decisions regarding whether or not transformations such as differencing were necessary before more extensive econometric examination [34–36].

After confirming stationarity, three econometric methods were employed to examine the relationships among the variables: Granger causality to identify linear predictive connections, quantile causality to capture nonlinear and asymmetric effects across different parts of the distribution, and wavelet coherence to analyze time-varying co-movements across multiple time scales. Together, these methods provide a more comprehensive framework to understand the dynamic and complex interactions relevant to the research context.

Wavelet coherence and quantile causality were preferred over conventional techniques due to their superior capacity to reflect the complex and time-dependent nature of relationships in markets such as tourism and clean energy. Traditional approaches often rely on average effects and assume stable relationships, which may overlook the actual dynamics of these sectors. Wavelet coherence captures co-movements in both time and frequency domains, highlighting how short- and long-term interactions evolve over time. Meanwhile, quantile causality reveals causal effects that may only be present during extreme or volatile periods by evaluating relationships across various quantiles. These advantages make both methods highly suitable for uncovering heterogeneous and time-varying patterns in the data that traditional techniques may fail to detect.

## Augmented Dickey-Fuller (ADF) test

The Augmented Dickey-Fuller (ADF) test serves as an extension of the standard Dickey-Fuller test, designed to accommodate more complex stochastic processes by incorporating lagged differences of the dependent variable. It assesses the null hypothesis that a unit root is present in a time series against the alternative hypothesis of stationarity. The ADF test specification is presented in Equation 1 [34].

$$\Delta Y_t = \alpha + \beta t + \gamma Y_{t-1} + \sum_{i=1}^{n} \delta_i \Delta Y_{t-i} + \varepsilon_t \tag{1}$$

where $\Delta Y_t$ represents the first difference of the series, t is a deterministic trend component, and $\varepsilon_t$ is a white noise error term.

Null Hyppthesis ($H_0$): $\gamma=0$, indicating the presence of a unit root (the series is non-stationary)

Alternative Hypothesis ($H_a$): $\gamma \neq 0$, indicating that the series is stationary.

The coefficient $\gamma$ is of particular importance; if it is found to be significantly different from zero, the null hypothesis of a unit root is rejected, indicating that the series is stationary. The selection of lag length in the ADF test is crucial, as it accounts for serial correlation and ensures robust estimation [34].

## Phillips-Perron (PP) test

The Phillips-Perron (PP) test, introduced by Phillips and Perron serves as an alternative to the ADF test, addressing potential issues related to heteroskedasticity and autocorrelation in the error terms. Unlike the ADF test, which explicitly includes lagged differences to correct for serial correlation, the PP test nonparametrically adjusts the test statistics using the Newey-West heteroskedasticity and autocorrelation consistent (HAC) covariance estimator. The test is based on the following regression model, as shown in Equation 2 [35].

$$\Delta Y_t = \alpha + \beta t + \gamma Y_{t-1} + \varepsilon_t \tag{2}$$

The null and alternative hypotheses of the PP test are identical to those of the ADF test:

Ho: $\gamma=0$, The time series contains a unit root (non-stationary).

 

$H_1$: $\gamma \neq 0$, The time series is stationary.

The advantage of the PP test lies in its ability to provide robust estimates in the presence of structural breaks or heteroskedasticity, making it a preferred choice in certain empirical applications [35].

**Kwiatkowski-Phillips-Schmidt-Shin (KPSS) test**

The Kwiatkowski-Phillips-Schmidt-Shin (KPSS) test, proposed by Kwiatkowski et al. [36], differs fundamentally from the ADF and PP tests, as it reverses the null and alternative hypotheses. In the KPSS framework, the null hypothesis assumes stationarity, while the alternative hypothesis suggests the presence of a unit root.
$H_0$: $Y_t = \beta_0 + \beta_1 t + \varepsilon_t$ The time series is stationary.
$H_1$: $Y_t = \beta_0 + \beta_1 t + rt + \varepsilon_t$ The time series is non-stationary (contains a unit root).
   The test is based on decomposing the time series into a deterministic trend, a random walk, and a stationary error term. By assessing the variance of the random walk component, the KPSS test determines whether the series deviates significantly from stationarity. The model can be expressed as follows:

$$Y_t = \beta_0 + \beta_1 t + rt + \varepsilon_t \tag{3}$$

Where:
   $Y_t$:Observed Time Series
   t:time trend
   $rt$: random trend component
   As given in Equation (3), the KPSS model accounts for both a deterministic trend and a stochastic component, enabling the test to assess whether the time series is stationary around a deterministic trend. If the test statistic exceeds the critical values, the null hypothesis is rejected, indicating that the series is likely to be non-stationary [36]. Compared to the ADF and PP tests, the KPSS test provides complementary insights, often used in conjunction with unit root tests to ensure robustness in stationarity assessment [36].

**Fourier stationarity test**

The Fourier stationarity test extends conventional unit root tests by incorporating a flexible nonlinear trend modeled via trigonometric Fourier functions. This methodology allows for the approximation of structural breaks, cyclic behaviors, and smooth transitions without imposing exogenous breakpoints [37]. The null and alternative hypotheses in the Fourier stationarity test are as follows:

- $H_0$: The time series contains a unit root (non-stationary).

- $H_1$: The time series is stationary.

The test model can be represented as follows:

$$\Delta Y_t = \alpha + \beta t + \Sigma \left[ \gamma_k \sin (2\pi kt/T) + \gamma_k \cos (2\pi kt/T) \right] + \rho Y_{t-1} + \varepsilon_t \tag{4}$$

where:

- α represents the intercept,

- βt captures the linear trend component,

- $\gamma_k$ and $\delta_k$ are Fourier coefficients modeling periodic fluctuations,

- k is the number of Fourier frequencies,

- T is the sample size, and

- $\rho Y_{t-1}$ accounts for the presence of a unit root.

As given in Equation (4), The Fourier approach is advantageous because it enables the detection of structural changes without prior knowledge of their location, unlike conventional breakpoint tests that require ex-ante specification [37]. This flexibility enhances the robustness of stationarity testing in cases where macroeconomic and financial data exhibit nonlinear periodicities.

**Quantile augmented Dickey-Fuller (QADF) test**

The Quantile Augmented Dickey-Fuller (QADF) test, introduced by Koenker and Xiao is an extension of the traditional Augmented Dickey-Fuller (ADF) test that examines the stationarity properties of a time series across different quantiles. Unlike the conventional ADF test, which focuses on the mean behavior of a series, the QADF test investigates the unit root hypothesis across various quantiles of the conditional distribution, providing a more detailed and robust analysis of stationarity [38]. The QADF test is instrumental when analyzing time series that exhibit asymmetric dynamics, heavy tails, or conditional heteroskedasticity, as it allows for investigating nonlinear adjustments and heterogeneous persistence at different distribution points.

This makes the QADF test more robust to distributional irregularities than the standard ADF test.

The model for the QADF test is formulated as shown in Equation 5:

$$Q_\tau \left( \Delta Y_t / X_t \right) = \alpha_\tau + \beta_\tau t + \gamma_t Y_{t-1} + \sum_{i=1}^{n} \delta_{i,\tau} \Delta Y_{t-i} + \varepsilon_{t,\tau} \tag{5}$$

where:

$Q_\tau \left( \Delta Y_t / X_t \right)$ denotes e conditional quantile of $\Delta Y_t$ given the information set Xt.

$\tau$ is the quantile level, indicating the point of the conditional distribution being analyzed.

$\gamma_\tau$ represents the quantile-specific unit root coefficient.

$\varepsilon_{t,\tau}$ is the quantile-specific error term.

The hypotheses for the QADF test are defined as follows:

- $H_0$: $\gamma_\tau = 0$, $\forall \tau \epsilon (0,1)$-The time series contains a unit root at a given quantile (non-stationary at that quantile).

- $H_1$: $\gamma_\tau < 0$, *for at least one* $\tau$− The time series is stationary at the specified quantile.

The rejection of the null hypothesis indicates that the shock effects are quantile-dependent, showing nonlinear adjustments and asymmetric Dynamics [38].

**Granger causality**

Granger causality is a widely used econometric method for assessing whether one time series can predict another [39]. The fundamental idea is that if a variable X provides statistically significant information about the future values of another variable Y, then X is said to Granger-cause Y. The test is based on the following vector autoregressive (VAR) model:

$$Y_t = \alpha + \sum \beta_i Y_{t-i} + \sum \gamma_i X_{t-i} + \varepsilon_t \tag{6}$$

where:

- α is the intercept term,

- $\beta_i$ and $\gamma_i$ represent the lagged coefficients,

- $\varepsilon_t$ is the error term.

As given in Equation (6), the model examines whether the inclusion of lagged values of X significantly improves the prediction of Y
    The null hypothesis of no causality is given by

- $H_0$: $\gamma_i = 0$ for all i (X does not Granger-cause Y).

- $H_1$: $\gamma_i \neq 0$ for at least one i (X Granger-causes Y).

If the null hypothesis is rejected, it suggests that past values of X significantly improve the prediction of Y. While Granger causality is a powerful tool, it is limited in its ability to capture nonlinear dependencies and quantile-specific causality effects

## Quantile causality

The quantile causality framework extends the traditional Granger causality test by examining causal effects at different quantiles of the conditional distribution of the dependent variable [40,41]. This approach is particularly useful when relationships between variables vary across different levels of economic activity or financial market conditions. The quantile causality test is based on the following regression model, as shown in Equation 7:

$$Q_t(\tau) = \alpha(\tau) + \Sigma \beta_i(\tau)\, Y_{t-i} + \Sigma \gamma_i(\tau)\, X_{t-i} + \varepsilon_t(\tau) \qquad (7)$$

where:

- $Q_t(\tau)$ represents the $\tau$-th quantile of the conditional distribution of Y,

- $\alpha(\tau)$ is the intercept at quantile $\tau$,

- $\beta_i(\tau)$ and $\gamma_i(\tau)$ represent the quantile-specific lagged coefficients,

- $\varepsilon_t(\tau)$ is the quantile-specific error term.

The null hypothesis is formulated as follows:

- $H_o$: $\gamma_i(\tau) = 0$ for all i (No quantile-specific causality from X to Y).

- $H_l$: $\gamma_i(\tau) \neq 0$ for at least one i (Quantile causality exists at $\tau$-th quantile).

Unlike traditional Granger causality, which provides a single inference over the entire sample, the quantile causality test allows researchers to assess whether causality is present in extreme conditions, such as financial crises or economic booms [40].

## Structural Var

Structural Vector Autoregression (SVAR) models extend the reduced-form VAR framework by imposing theory-based identification restrictions to uncover structural shocks. The reduced-form VAR model can be written as:

$$Y_t = B_1 Y_{t-1} + \cdots + B Y_{t-p} + u \qquad (8)$$

where $Y_t$ is a vector of endogenous variables and $u_t$ is a vector of residuals that may be contemporaneously correlated. To identify structural innovations, the SVAR representation takes the form:

$$A_o \, Y_t \; = \; A_1 \, Y_{t-1} \; + \; \cdots \; + \; A_p \, Y_t \; + \; \varepsilon \tag{9}$$

with $\varepsilon_t$ denoting structural shocks such that $E[\varepsilon_t \, \varepsilon_t'] = I$, and $A_0 u_t = \varepsilon_t$. Identification requires sufficient restrictions on $A_0$, typically derived from economic theory through short-run [42], long-run [43], or sign restrictions [44]. This framework enables causal inference in multivariate time series analysis

## Wavelet-coherence analysis

Wavelet Coherence Analysis is a powerful method for examining the time-frequency relationships between two non-stationary signals.

The foundation of wavelet coherence lies in the continuous wavelet transform (CWT), which decomposes a time series into time-frequency space using wavelets functions that are localized in both time and frequency.

Wavelet coherence is computed as the normalized cross wavelet transform:

$$R^2_{xy}(a,b) = \frac{\left| S\left(b^{-1} W_{xy}(a,b)\right) \right|^2}{S(b^{-1} W_x(a,b))^2 \, S(b^{-1} W_y(a,b))^2} \tag{10}$$

Here, S denotes a smoothing operator in both time and scale [45].The resulting wavelet coherence coefficient, $R^2(a, b)$, ranges between 0 and 1, indicating the degree of linear correlation between the signals at each time-frequency point. Values close to 1 suggest a strong correlation, while values near 0 indicate little to no correlation [46].

The phase difference in the function as;

$$\phi_{xy} = arctan\left( \frac{Im\left[ S\left(b^{-1} W_{xy}(a,b)\right)\right]}{RE\left[ S\left(b^{-1} W_{xy}(a,b)\right)\right]} \right), \; with \; \phi_{xy} \, \epsilon [-\pi, \pi], \tag{11}$$

As shown in Equations (10) and (11), the imaginary (Im) and real (Re) components of the smoothed power spectrum represent different aspects of the signal's characteristics. Specifically, the phase angle φxy\phi{xy}φxy offers valuable information about the correlation and potential cause-and-effect relationship (lead-lag dynamics) between two variables. In the context of the wavelet power spectrum, arrows pointing to the right ($\rightarrow$) or left ($\leftarrow$) illustrate whether the variables are positively correlated (in-phase) or negatively correlated (out-of-phase), respectively. Furthermore, an upward-right diagonal arrow ($\nearrow$) suggests that x(t)x(t)x(t) precedes y(t)y(t)y(t), indicating a lead relationship, whereas a downward-left diagonal arrow ($\swarrow$) indicates that y(t)y(t)y(t) precedes x(t)x(t)x(t), implying a lag relationship [45].

## 4. Emprical results

At the beginning of the analysis, the Augmented Dickey-Fuller (ADF), Phillips-Perron (PP), and Kwiatkowski-Phillips-Schmidt-Shin (KPSS) unit root tests were applied to examine the stationarity of the series. The test results obtained are presented in Table 3.

Table 3 shows that all series become stationary at the 1%, 5%, and 10% significance levels after first differencing, based on the Augmented Dickey-Fuller (ADF) and Phillips-Perron (PP) tests.

Regarding the KPSS test results, the LWILDER series was non-stationary at the 10% significance level in the constant model. However, in the constant and trend model, the series was stationary at all significance levels. Nevertheless, given that the computed KPSS value of 0.231 is close to the critical values at the 1%, 5%, and 10% significance levels, which are 0.216, 0.146, and 0.119, respectively, and considering the constant-estimated test values, as well as the results from the ADF and Phillips-Perron tests, it was decided to analyze by taking the first difference of the relevant series.

Accordingly, the KPSS test, similar to the ADF and Phillips-Perron tests, concluded that all series were I(1). The test results, as presented in the table, are statistically significant.

**Table 3. Conventional Analysis of the unit root.**

| Approach | ADF | | | | |
|---|---|---|---|---|---|
| | **Level** | | **First Diff** | | |
| **Variables** | **Constant** | **Constant and Trend** | **Constant** | **Constant and Trend** | **RESULT** |
| LWILDER | −1.872 | −2.133 | −9.449*** | −9.431*** | I(1) |
| LSTOXXR | −2.080 | −2.086 | −13.193*** | −13.209*** | I(1) |
| LDJUSTT | −0.346 | −2.703 | −13.675*** | −14.184*** | I(1) |
| LFTXIN | −2.089 | −2.196 | −13.536*** | −13.494*** | I(1) |
| LAXHRKD | −1.482 | −2.541 | −14.360*** | −14.346*** | I(1) |
| Approach | **PP** | | | | |
| | Level | | First Diff | | |
| | Constant | Constant and Trend | Constant | Constant and Trend | RESULT |
| LWILDER | −1.628 | −1.795 | −9.531*** | −9.506*** | I(1) |
| LSTOXXR | −2.104 | −2.185 | −13.198*** | −13.213*** | I(1) |
| LDJUSTT | −0.346 | −2.658 | −13.675*** | −14.253*** | I(1) |
| LFTXIN | −2.089 | −2.196 | −13.554*** | −13.511*** | I(1) |
| LAXHRKD | −1.421 | −2.430 | −14.444*** | −14.432*** | I(1) |
| Approach | **KPSS** | | | | |
| | Level | | First Diff | | |
| | KPSS Test Statistic | | KPSS Test Statistic | | |
| | Constant | Constant and Trend | Constant | Constant and Trend | RESULT |
| LWILDER | 0.395 * | 0.231*** | 0.151 | 0.132 | I(1) |
| LSTOXXR | 0.984*** | 0.337*** | 0.119 | 0.037 | I(1) |
| LDJUSTT | 0.926*** | 0.258*** | 0.658 | 0.073 | I(1) |
| LFTXIN | 0.312*** | 0.200* | 0.064 | 0.058 | I(1) |
| LAXHRKD | 1.460*** | 0.279*** | 0.081 | 0.03 | I(1) |

Significance levels: * indicates significance at the 10% level, ** at the 5% level, and *** at the 1% level. Note: For ADF, significance indicates rejection of the unit root (i.e., the series is stationary); or KPSS, significance implies rejection of stationarity (i.e., the series is non-stationary).

According to the results of traditional unit root tests, none of the LWILDER, LSTOXXR, LDJUST, LFTXIN, and LAXHRKD series were found to be stationary at level, whereas they all became stationary after first differencing. Following this finding, the Fourier unit root test was conducted, providing a more robust analysis compared to traditional unit root tests while accounting for nonlinear trends and incorporating unknown structural breaks into the process. The results are presented in Table 4.

The results of the Fourier unit root test indicate the presence of a unit root in all the variables analyzed. For none of the variables did the Fourier values exceed the critical values (1%: −3.99, 5%: −3.43, 0.10%: −3.13) leading to the conclusion that all series are non-stationary. These findings suggest that the series contain long-term trends or structural breaks. Therefore, the first differences of the series were taken, and the test was reapplied. The results are presented in the Table 5.

After identifying non-stationarity in the levels of all variables using the Fourier unit root test, the first differences were calculated and retested. The results indicate that the first differences are stationary for all variables, with optimal Fourier values significantly exceeding the critical values at the 1% level. This suggests that all series are integrated of order one, I(1).

Following the application of the Fourier unit root test, the Quantile-ADF test was conducted for all series to analyze stationarity across different quantiles. The results are presented in Table 6 and Table 7.

**Table 4. Fourier Statistic Results.**

| Variable | Optimal k | Fourier Value |
|---|---|---|
| LWILDER | 1 | −2.2603*** |
| LFTXIN | 2 | −2.5359*** |
| LDJUSTT | 1 | −1.4444*** |
| LSTOXXR | 1 | −2.2897*** |
| LAXHRKD | 1 | −1.8093*** |

Significance levels: * indicates significance at the 10% level, ** at the 5% level, and *** at the 1% level.

**Table 5. Fourier unit root test I(1).**

| Variable | Optimal k | Fourier Value |
|---|---|---|
| DLWILDER | 4 | −8.7569*** |
| DLFTXIN | 2 | −9.8895*** |
| DLDJUSTT | 2 | −10.1885*** |
| DLSTOXXR | 5 | −9.642*** |
| DLAXHRKD | 5 | −10.9218*** |

**Table 6. Quantile ADF Results(Constant).**

| VARIABLES | QUANTILES | | | | | | | | | |
|---|---|---|---|---|---|---|---|---|---|---|
| | 0.1 | | 0.25 | | 0.5 | | 0.75 | | 0.9 | |
| | t-statistics | p value | t-statistics | p value | t-statistics | p value | t-statistics | p value | t-statistics | p value |
| LWILDER | −0.468 | 0.976 | −0.915 | 0.939 | −2.339 | 0.437 | −2.687 | 0.291 | −2.394 | 0.412 |
| LFTXIN | −2.114 | 0.529 | −0.528 | 0.976 | −0.613 | 0.974 | −2.432 | 0.397 | −1.906 | 0.616 |
| LDJUSTT | −4.133** | 0.019 | −2.364 | 0.43 | −2.139 | 0.519 | −1.962 | 0.592 | −2.794 | 0.246 |
| LSTOXXR | −0.91 | 0.934 | −1.503 | 0.77 | −2.824 | 0.238 | −2.107 | 0.532 | −2.108 | 0.532 |
| LAXHRKD | −1.867 | 0.623 | −1.052 | 0.918 | −1.753 | 0.677 | −2.096 | 0.536 | −1.803 | 0.658 |
| | | | | | | | | | | |

**Table 7. Quantile ADF Results(Constant+Trend).**

| VARIABLES | QUANTILES | | | | | | | | | |
|---|---|---|---|---|---|---|---|---|---|---|
| | 0.1 | | 0.25 | | 0.5 | | 0.75 | | 0.9 | |
| | t-statistics | p value | t-statistics | p value | t-statistics | p value | t-statistics | p value | t-statistics | p value |
| LWILDER | −1.226 | 0.881 | −1.439 | 0.850 | −2.103 | 0.726 | −2.712 | 0.576 | −2.309 | 0.679 |
| LFTXIN | −2.221 | 0.702 | −1.142 | 0.892 | −0.884 | 0.920 | −2.442 | 0.646 | −2.123 | 0.722 |
| LDJUSTT | −2.180 | 0.710 | −3.533 | 0.363 | −2.058 | 0.737 | −2.030 | 0.743 | −2.232 | 0.699 |
| LSTOXXR | −2.190 | 0.708 | −1.644 | 0.818 | −2.443 | 0.646 | −2.027 | 0.743 | −2.006 | 0.747 |
| LAXHRKD | −1.961 | 0.758 | −0.951 | 0.913 | −2.276 | 0.688 | −2.365 | 0.666 | −2.365 | 0.666 |

Significance levels: * indicates significance at the 10% level, ** at the 5% level, and *** at the 1% level.

The results presented in Table 6 indicate that the general trend across all series suggests non-stationarity at the level for all quantiles, with the only exception being the first quantile of the LDJUSTT series, which appears to be stationary at the 5% significance level. When the results of Table 6 are evaluated, applying quantile ADF tests to both constant and

trend models reveals that only the LDJUSTT series demonstrates stationarity at the 0.25 quantile, whereas no stationarity is observed for the other series and quantiles. Consequently, the first differences of the series were taken, and stationarity tests were conducted accordingly, confirming the persistence of non-stationarity across most series and quantiles.

The results presented in Table 8 and Table 9 indicate that the first differences of all series are non-stationary at the 0.10 quantile but become stationary at the 0.25, 0.50, 0.75, and 0.90 quantiles. The fact that all series are stationary at the 0.50 quantile, with only the smallest quantile (0.10) showing non-stationarity, leads to the conclusion that the series are generally stationary at the first difference level.

Following this stage, in order to determine the relationship between the LWILDER index and the other indices, the Pairwise Causality Test was conducted, and the results are presented in Table 10.

According to the results of the traditional causality test presented in Table 10, the WILDER index does not cause the FTXIN, DJUSTT, STOXX, and AXHRKD series. All p-values exceed the 5% significance threshold, indicating that the

**Table 8. First difference Quantile ADF Results (Constant).**

| VARIABLES | QUANTILES | | | | | | | | | |
|---|---|---|---|---|---|---|---|---|---|---|
| | 0.1 | | 0.25 | | 0.5 | | 0.75 | | 0.9 | |
| | t-statistics | p value | t-statistics | p value | t-statistics | p value | t-statistics | p value | t-statistics | p value |
| DLWILDER | −1.555 | 0.742 | −2.606 | 0.335 | −3.621 | 0.037** | −3.33 | 0.07* | −3.309 | 0.072* |
| DLFTXIN | −2.432 | 0.408 | −3.615 | 0.044** | −4.296 | 0.01*** | −5.058 | 0.01*** | −5.548 | 0.01*** |
| DLDJUSTT | −2.352 | 0.438 | −4.126 | 0.014*** | −2.994 | 0.168 | −3.813 | 0.021** | −3.212 | 0.089* |
| DLSTOXXR | −1.612 | 0.72 | −3.804 | 0.029*** | −4.359 | 0.01*** | −6.012 | 0.01*** | −4.924 | 0.01*** |
| DLAXHRKD | −1.835 | 0.635 | −3.429 | 0.066** | −4.041 | 0.012** | −5.617 | 0.01*** | −5.313 | 0.01*** |

Significance levels: * indicates significance at the 10% level, ** at the 5% level, and *** at the 1% level.

**Table 9. First difference Quantile ADF Results (Constant+Trend).**

| VARIABLES | QUANTILES | | | | | | | | | |
|---|---|---|---|---|---|---|---|---|---|---|
| | 0.1 | | 0.25 | | 0.5 | | 0.75 | | 0.9 | |
| | t-statistics | p value | t-statistics | p value | t-statistics | p value | t-statistics | p value | t-statistics | p value |
| DLWILDER | −3.137 | 0.463 | −4.808 | 0.100* | −6.719 | 0.007*** | −6.357 | 0.012*** | −7.973 | 0.000*** |
| DLFTXIN | −3.044 | 0.49 | −4.539 | 0.100* | −6.255 | 0.015*** | −7.45 | 0.001*** | −8.596 | 0.000*** |
| DLDJUSTT | −2.825 | 0.547 | −3.916 | 0.271 | −4.809 | 0.100* | −6.04 | 0.022** | −6.336 | 0.012*** |
| DLSTOXXR | −2.314 | 0.678 | −4.434 | 0.172 | −6.659 | 0.008*** | −6.788 | 0.006*** | −7.195 | 0.003*** |
| DLAXHRKD | −2.711 | 0.576 | −5.047 | 0.088** | −6.461 | 0.010*** | −7.742 | 0.001*** | −8.019 | 0.000*** |

Significance levels: * indicates significance at the 10% level, ** at the 5% level, and *** at the 1% level.

**Table 10. Pairwise Granger Causality Test Results.**

| PAIRWISE GRANGER CAUSALITY TESTS | F value | Prob |
|---|---|---|
| WILDER IS NOT THE GRANGER CAUSE DLFTXN | 0.002 | 0.962 |
| WILDER IS NOT THE GRANGER CAUSE DJUSST | 0.015 | 0.985 |
| WILDER IS NOT THE GRANGER CAUSE STOXXR | 0.102 | 0.903 |
| WILDER IS NOT THE GRANGER CAUSE AXHRKD | 0.729 | 0.484 |

Significance levels: * indicates significance at the 10% level, ** at the 5% level, and *** at the 1% level.

clean energy index does not exhibit any statistically significant linear predictive relationship with the tourism indices in the Granger causality framework.

However, the Granger test captures only average effects under the assumption of linearity and time-invariance. To enhance the sensitivity of the analysis and uncover potential heterogeneous causal relationships that may vary across different market conditions, the Quantile Causality Test was conducted between the WILDER series and each of the tourism-related indices. The quantile regression estimates rely on Huber sandwich standard errors and kernel-based sparsity estimation. As the model does not implement bootstrap-based inference, the reported p-values may not fully capture the distributional uncertainty under serial dependence or heteroskedasticity. This limitation is acknowledged when interpreting the results. This method allows for a more comprehensive exploration of causality by examining how the relationship behaves across different points of the conditional distribution of the dependent variable, particularly under extreme or crisis conditions. The results are presented in Table 11.

The results of the quantile causality analysis between the DWILDER index and the DFTXIN, DJUSTT, DLSTOXX, and DLAXHRKD series, as presented in Table 11, provide important insights into how the clean energy sector interacts with tourism indices under varying market conditions. For the DWILDER–DFTXIN relationship, significant causality is detected at the 0.5 and 0.75 quantiles. This implies that changes in the clean energy index have a predictive influence on the Chinese tourism sector primarily under normal to moderately strong market conditions. Regarding the DWILDER–DLDJUSTT pair, causality is found at the extreme lower and upper quantiles (0.05, 0.10, 0.90, 0.95), as well as at the 0.75 quantile. These results suggest that the impact of clean energy movements on the U.S. tourism sector is asymmetric and becomes especially pronounced during both market downturns and booms, highlighting the presence of nonlinear dynamics. In the case of DWILDER and DLSTOXXR, a statistically significant causality is identified only at the 0.05 quantile, indicating that during adverse or bearish periods, changes in the clean energy index may play a role in explaining movements in the European tourism index. Lastly, the causality from DWILDER to DLAXHRKD is observed at the lower quantiles (0.05, 0.10, and 0.25), reflecting that under weak or stressful market conditions, the Australian tourism sector is particularly responsive to variations in clean energy activity. Notably, the absence of causality at higher quantiles (e.g., 0.75) further supports the view that these interactions are condition-dependent rather than uniform.

Overall, the findings confirm that the causal relationships between clean energy and tourism indices are not constant across the distribution, emphasizing the importance of using a quantile-based framework to uncover hidden or asymmetric dependencies that traditional mean-based tests may fail to detect.

A more nuanced understanding of these asymmetric patterns may be achieved by situating them within the theoretical perspectives of behavioral finance and portfolio allocation. Under lower quantiles (e.g., 0.05, 0.10), market stress or pessimistic sentiment may prompt investors to reallocate capital between sectors based on perceived risk and ESG alignment,

**Table 11. Checks for Granger-Causality in quantiles of the Wılder and tourism sectors.**

|  | DWILDER to DLFTXIN | | DWILDER to DLDJUSST | | DWILDER to DLSTOXXR | | DWILDER to DLAXHRKD | |
|---|---|---|---|---|---|---|---|---|
|  | Wald | p | Wald | p | Wald | p | Wald | P |
| **0.05** | 1.216 | 0.270 | 24.715 | 0.000*** | 17.415 | 0.000*** | 41.085 | 0.000*** |
| **0.1** | 0.258 | 0.611 | 11.812 | 0.000*** | 0.251 | 0.616 | 41.222 | 0.000*** |
| **0.25** | 0.027 | 0.869 | 1.599 | 0.206 | 0.863 | 0.353 | 34.430 | 0.000 |
| **0.5** | 5.71 | 0.010*** | 0.013 | 0.908 | 0.018 | 0.895 | 1.975 | 0.159*** |
| **0.75** | 23.568 | 0.000*** | 2.208 | 0.100* | 0.021 | 0.886 | 2.191 | 0.139 |
| **0.9** | 0.375 | 0.540 | 10.314 | 0.001*** | 0.044 | 0.835 | 0.029 | 0.865 |
| **0.95** | 0.400 | 0.527 | 25.837 | 0.000*** | 1.042 | 0.307 | 0.962 | 0.327 |

Significance levels: * indicates significance at the 10% level, ** at the 5% level, and *** at the 1% level.

making the tourism sector more sensitive to clean energy shocks. Conversely, during upper quantiles (e.g., 0.90, 0.95), booming market conditions may amplify capital flows into both sectors, particularly in economies with strong sustainability agendas. The absence of causality at central quantiles in some regions may reflect a lack of sectoral interdependence under stable conditions. These mechanisms help contextualize the distribution-dependent causal relationships observed in the analysis.

Following the quantile Granger causality results, Fig 1 visually summarizes the significant causal effects observed across quantiles and regional tourism indices. This allows for easier identification of how the influence of clean energy varies under different market scenarios.

Heatmap illustrating statistically significant quantile Granger causality relationships ($p < 0.05$) from the clean energy index (DWILDER) to regional tourism indices. Each cell marked as "1" denotes a significant causal effect at the corresponding quantile and region, while "0" indicates no significance.

Fig 1 demonstrates that the strength and pattern of causality vary across both quantile levels and regional markets, underscoring the heterogeneous and asymmetric nature of the energy–tourism nexus. The United States displays consistent significance at both lower and upper quantiles, suggesting that clean energy developments affect tourism during periods of both economic contraction and expansion. Australia shows significance primarily in lower quantiles, indicating a stronger sensitivity during adverse conditions. In contrast, Europe exhibits significance only at the lowest quantile (0.05), pointing to vulnerability during extreme downturns. Meanwhile, China shows responsiveness at the median and upper-middle quantiles, reflecting a more growth-oriented linkage. These results reinforce the importance of quantile-based approaches in capturing non-linear and distribution-dependent causal dynamics that conventional mean-based methods may overlook.

Table 12 summarizes the results of the Lagrange Multiplier (LM) serial correlation tests conducted for each SVAR model, while Fig 2 presents the inverse roots of the AR characteristic polynomial. The LM test assesses whether the residuals exhibit autocorrelation; p-values above the 0.05 threshold across all lag orders indicate that the residuals are serially

**Fig 1. Significant quantile causality heatmap across regions.**

**Table 12. LM Test Results Indicating Residual Autocorrelation Status Across Models.**

| Model | Lag Range | Min p value | Max p value | Result |
|---|---|---|---|---|
| DLWILDER→ DLFTXIN | 1-12 | 0.218 | 0.926 | No Autocorrelation |
| DLWILDER→ DLDJUSTT | 1-3 | 0.356 | 0.721 | No Autocorrelation |
| DLWILDER→ DLSTOXXR | 1-12 | 0.378 | 0.977 | No Autocorrelation |
| DLWILDER →DLAXHRKD | 1-12 | 0.194 | 0.944 | No Autocorrelation |

Appendix Figure A1. Inverse Roots of AR Characteristic Polynomial for Each Model

AR Roots – China

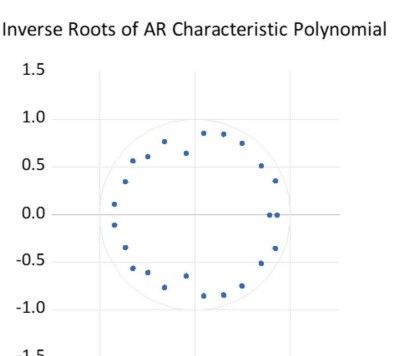

AR Roots – USA

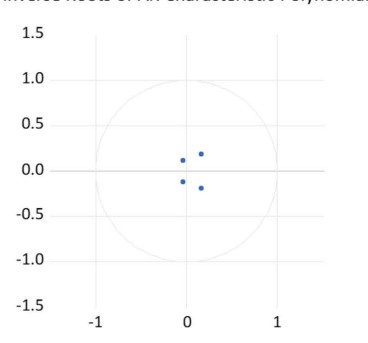

AR Roots – Europe

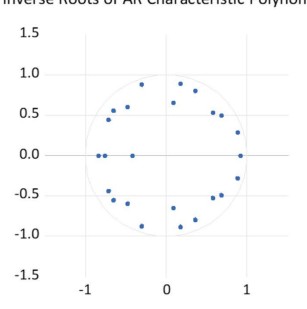

AR Roots – Australia

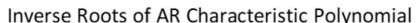
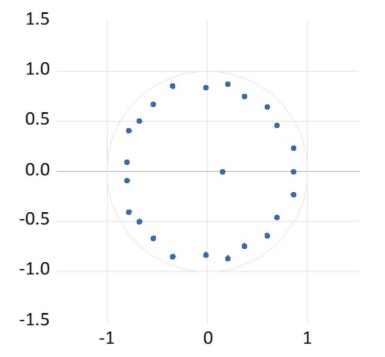

**Fig 2. Inverse Roots of the AR Characteristic Polynomial for SVAR Models.**

uncorrelated, supporting model adequacy. Meanwhile, the stability of each SVAR specification is evaluated based on the location of the characteristic roots. When all roots lie strictly within the unit circle, the system is considered dynamically stable, validating the reliability of the impulse response functions. Taken together, the results in Table 1 and Fig 1 confirm the statistical soundness and robustness of the estimated SVAR models.

Based on the SVAR estimates presented in Table 13, the dynamic responses of tourism sector indices to shocks in the clean energy index (DLWILDER) display substantial regional heterogeneity, both in terms of sign and statistical significance across time horizons.

**Table 13. Structural Vector Autoregression (SVAR) Results.**

| Shock Direction | Horizon | Coefficient | p-Value |
|---|---|---|---|
| DLWILDER→ DLFTXIN | C1 | 0.28 | 0.00 |
| | C2 | 0.08 | 0.00 |
| | C3 | 0.09 | 0.00 |
| DLWILDER→ DLDJUSTT | Horizon | Coefficient | p-Value |
| | C1 | −0.02 | 0.80 |
| | C2 | 0.08 | 0.00 |
| | C3 | 0.08 | 0.00 |
| DLWILDER→ DLSTOXXR | Horizon | Coefficient | p-Value |
| | C1 | −0.11 | 0.06 |
| | C2 | 0.01 | 0.00 |
| | C3 | 0.01 | 0.00 |
| DLWILDER →DLAXHRKD | Horizon | Coefficient | p-Value |
| | C1 | −0.31 | 0.00 |
| | C2 | 0.08 | 0.00 |
| | C3 | 0.05 | 0.00 |

For China, the clean energy index exerts a consistently positive and statistically significant effect on the tourism sector across all three horizons (C1, C2, and C3). This robust pattern implies a strong and immediate responsiveness of China's tourism market to developments in clean energy, possibly reflecting the country's integrated sustainability strategies and growing environmental awareness within the tourism sector.

In the United States, no statistically significant short-run effect is observed (C1, $p = 0.80$), suggesting a delayed transmission mechanism. However, clean energy shocks exhibit statistically significant and positive effects in the medium and long term (C2 and C3, $p = 0.00$), indicating that policy-driven clean energy transitions gradually translate into favorable structural outcomes for the tourism industry. This delayed yet positive response may be associated with long-term infrastructure adaptation and changing investor expectations.

Regarding Europe, the short-term effect of clean energy shocks on tourism appears weakly negative and marginally significant ($p = 0.06$), possibly reflecting transitional frictions or sectoral inertia. Nevertheless, the medium- and long-run coefficients become positive and statistically significant, underscoring the adaptive capacity of the European tourism sector to leverage clean energy transformation over time. These findings suggest that while clean energy policies may introduce short-run uncertainties, they eventually foster more resilient and sustainable tourism dynamics within the region.

In the case of Australia, clean energy shocks generate a pronounced negative short-term effect (C1 = −0.31, $p = 0.00$), likely capturing adjustment costs, sectoral reallocation, or initial policy frictions. However, the direction of impact reverses over time, with medium- and long-term effects turning significantly positive. This trajectory implies that despite initial disruptions, clean energy transitions ultimately contribute to structural improvements in the Australian tourism sector, potentially through green innovation, improved destination image, or synergies with broader environmental policy goals.

Overall, these country-specific patterns reveal regionally heterogeneous yet economically consistent responses, with clean energy shocks generally yielding positive medium- to long-term impacts on tourism. This reinforces the context-dependent nature of energy-tourism dynamics and supports the robustness of the empirical framework.

According to Fig 3, the bootstrap-based impulse responses offer further insight into the dynamic linkages between clean energy shocks and regional tourism indices.

Appendix Figure A3. Bootstrap Impulse Response of Tourism Indices to Clean Energy Shocks

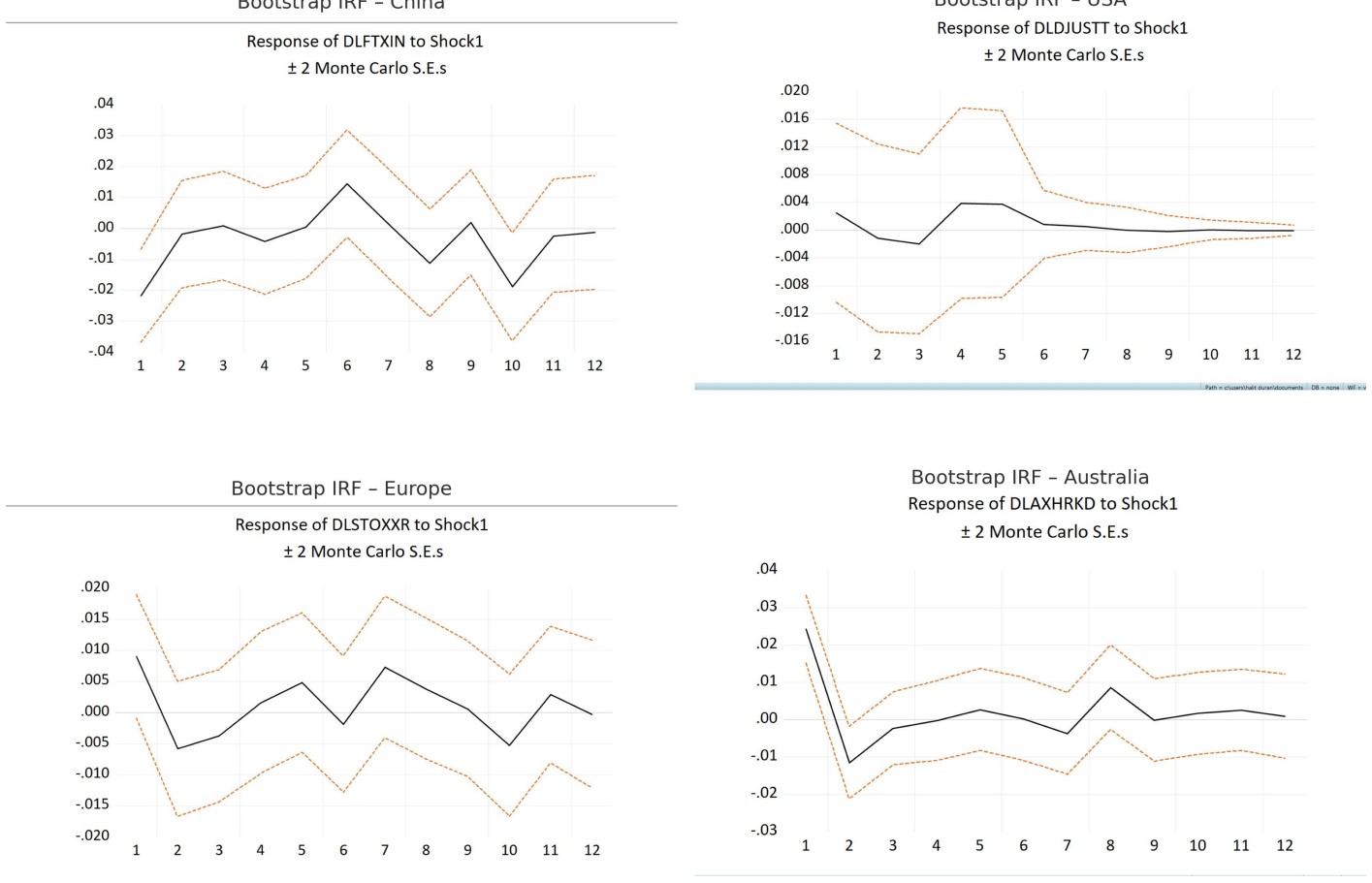

**Fig 3. Impulse Response of Tourism Indices to Clean Energy Shocks.** Note: Each panel presents the 12-month impulse response of regional tourism indices to a one standard deviation structural innovation in the clean energy index (DLWILDER), computed using Monte Carlo simulation with ±2 standard errors.

In the case of China, the results confirm a delayed yet positive effect, particularly around the sixth month. While statistical significance remains marginal, the overall consistency with asymptotic estimates enhances the credibility of this finding.

For the United States, the tourism sector appears largely unresponsive to clean energy shocks. The estimated effects remain close to zero, and the confidence bands indicate no statistically meaningful deviation over the forecast horizon.

In the European context, the responses fluctuate slightly but remain statistically insignificant across all periods. This outcome reinforces the limited transmission of clean energy shocks to the tourism sector, as previously observed.

As for Australia, the tourism index displays an initial negative response followed by a gradual shift toward a mildly positive trajectory. Although the responses fall within the error bands, the pattern may reflect short-run adjustment pressures followed by adaptation.

Collectively, the bootstrap-based impulse responses presented in Fig 3 validate the asymptotic results and highlight the heterogeneity in regional tourism sector sensitivities to clean energy shocks. While China demonstrates a relatively consistent and delayed positive effect, other regions such as the United States and Europe exhibit minimal or statistically insignificant responses, suggesting structural differences in sectoral exposure and policy responsiveness. Australia's pattern, though not statistically strong, points to an initial adjustment followed by adaptation. These findings underscore the importance of regional context in shaping the transmission of sustainability-oriented economic signals and support the use of distribution-sensitive approaches in policy-impact assessments.

In the final part of the study, wavelet analysis was conducted to examine the relationship between the DWILDER and the other series. In the wavelet figures, the 0–50 index corresponds to the period between 2010 and 2015, the 50–100 index covers 2016–2019, the 100–150 index corresponds to 2020–2022, and the 150–180 index represents the year 2023.

The color scale in the figures indicates the strength of wavelet coherence: warmer colors (such as red and orange) denote stronger co-movements between the two series, while cooler colors (e.g., blue and green) imply weaker or no significant relationship. Regions where coherence exceeds the 5% significance level are enclosed by thick boundary lines, indicating statistically reliable areas. The direction of the arrows reflects both correlation and lead-lag dynamics: rightward arrows indicate in-phase movement (positive correlation), while leftward arrows represent anti-phase behavior (negative correlation). Downward arrows indicate that the first series leads, and upward arrows indicate that the second series leads.

Wavelet coherence plots offer meaningful visual insights into time–frequency dependencies and potential lead–lag structures that may remain undetected by conventional time-domain methods. Although this study does not implement simulation-based significance testing and thus does not permit formal inference, the results provide an exploratory perspective on how the relationship between clean energy and tourism stock indices evolves over time. Following the methodological caveats highlighted by [47], the findings are interpreted with caution. Future research is encouraged to complement wavelet-based visualizations with simulation-supported tests to strengthen causal inference and improve analytical robustness.

Following the general framework outlined above, the wavelet coherence between DLWILDER and DLFTXIN is visualized in Fig 4.

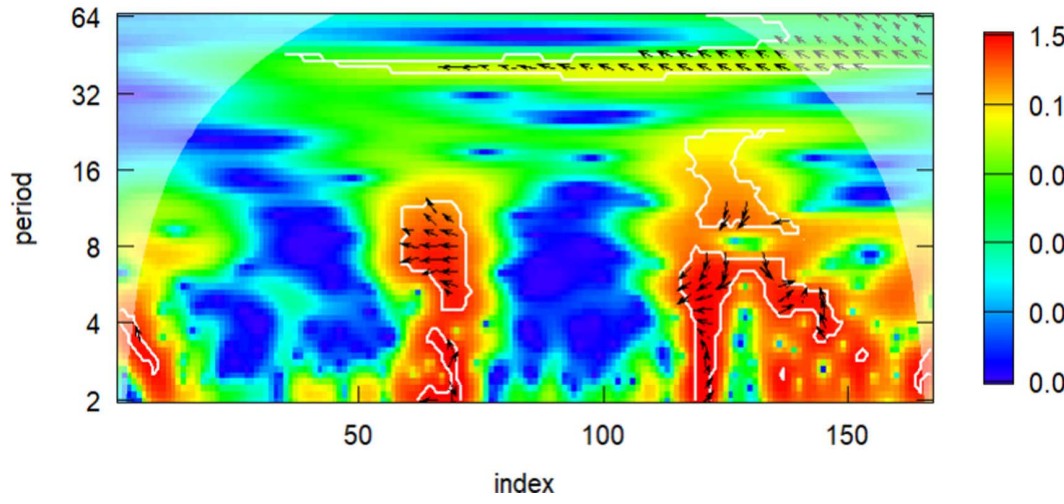

**Fig 4. Wavelet correlation between the DLWILDER and DLFTXIN.**

According to Fig 4, it can be observed that coherence is weak in the short term, particularly within the 2–4 month periods, with few visually distinguishable high-coherence regions. The period from 2016 to 2019 exhibits increased coherence in the medium frequencies (8–16 period), reflecting the impact of global economic stability and relatively balanced energy prices. The observed co-movement between the LWILDER and LFTXIN series may be associated with China's growing energy demand and the expansion in global trade volume, suggesting a potentially stronger linkage during this period. In long-term trends (32+months), the relationship is weak, with random fluctuations being predominant.

The wavelet coherence analysis between LWILDER and LFTXIN reveals intense medium-term economic cycles but weaker long-term relationships, highlighting the synchronous impacts of the COVID-19 pandemic and regional differences in economic recovery. The observed medium-frequency synchronization is particularly evident between 2020 and 2022, likely reflecting the joint impact of the pandemic and subsequent policy responses. In contrast, the weaker coherence in 2023 suggests diverging economic trajectories, with China's inward-oriented growth strategy reducing sensitivity to external energy price fluctuations. This decoupling may also reflect China's shifting policy focus toward domestic consumption and energy security in the post-pandemic era, supported by industrial policy initiatives such as the Dual Circulation Strategy. These findings offer exploratory insights into potential synchronization and divergence patterns between global tourism and energy-related activities.

The wavelet correlation analysis between the DLDWILDER and DLJUST variables is presented in Fig 5.

According to Fig 5, the wavelet coherence analysis between DLWILDER and DLDJUST exhibits strong visual co-movement in the medium-term (8–16 month) and long-term (32+month) cycles. During the 2016–2019 period, the observed coherence may reflect the heightened sensitivity of the U.S. tourism sector to global energy prices amid economic stability and rising demand. In contrast, the 2020–2022 period shows elevated coherence in medium frequencies, indicating synchronous economic shocks due to the COVID-19 pandemic, characterized by travel restrictions, declines in energy demand, and reduced consumer confidence. In the long term, a consistent co-movement pattern emerges, particularly after 2020, suggesting a possible lead-lag relationship that may reflect broader U.S. energy and macroeconomic conditions. However, in 2023, the low coherence and lagged phase differences reveal slower and delayed recovery patterns, reflecting uncertainties in energy prices and sluggish growth in U.S. tourism sector. This delayed recovery may be linked to the U.S.'s relatively limited reliance on green fiscal stimulus compared to other advanced economies, as well

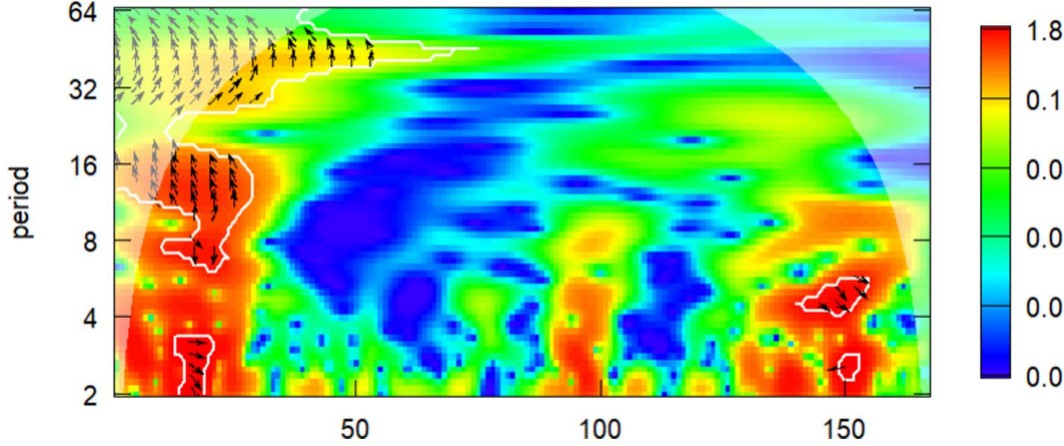

**Fig 5. Wavelet Correlation between the DLWILDER and DLDJUST.**

as the domestic orientation of its tourism sector. Moreover, policy fragmentation across states and a more market-driven approach to energy transition may have contributed to the observed lagged adjustments and weaker synchronization in the post-pandemic period.

As indicated by the phase arrows, a time-varying and directionally asymmetric relationship is observed. In particular, the presence of leftward and downward arrows in several key areas suggests that changes in the clean energy index typically precede and possibly influence movements in the U.S. tourism index.

These findings offer exploratory insights into the potential synchronization and divergence of tourism and energy market dynamics in the United States, especially when compared to faster post-pandemic recovery patterns in other regions. The results underscore the relevance of structural energy dependence in shaping regional economic resilience.

The wavelet correlation analysis between the DLDWILDER and DLSTOXX variables is presented in Fig 6.

According to Fig 6, the wavelet coherence analysis between DLWILDER and DLSTOXXR reveals a strong visual coherence pattern in the medium-term (8–16 month periods) and long-term (32 + month periods) cycles. During the 2016–2019 period, increased coherence may reflect Europe's heightened sensitivity to global energy prices amid economic stability and rising energy demand. In contrast, the 2020–2022 period shows high coherence in medium frequencies, indicating synchronous economic shocks due to the COVID-19 pandemic, characterized by travel restrictions, declines in energy demand, and reduced consumer confidence. In the long term, a consistent co-movement becomes evident, particularly after 2020, suggesting a possible lead-lag relationship that may reflect Europe's high energy dependence and macroeconomic cycles. However, in 2023, the low coherence and lagged phase differences reveal slower and delayed recovery patterns, possibly reflecting uncertainties in energy prices and sluggish growth in Europe's tourism sector. The relatively sluggish recovery and declining coherence observed in 2023 may be attributed to Europe's structural reliance on imported energy, compounded by the ongoing energy crisis triggered by the Russia–Ukraine conflict. In addition, the fragmented fiscal and tourism recovery strategies across EU member states, as well as delays in implementing large-scale green investments under the EU Recovery and Resilience Facility, may have weakened the transmission channel between clean energy and tourism sectors.

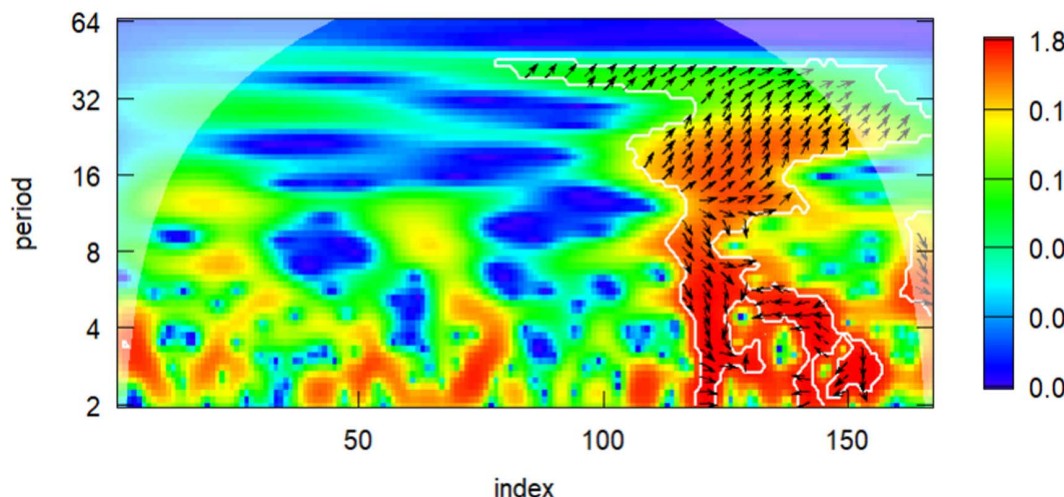

**Fig 6. Wavelet Correlation between the DLWILDER and DLSTOXX.**

The phase arrows reveal a time-varying and directionally asymmetric interaction. In several key regions, leftward and upward-left arrows suggest that changes in the clean energy index (DLWILDER) typically precede and may be associated with movements in the European tourism index (DLSTOXXR). These patterns imply delayed but meaningful transmission mechanisms across the sectors.

These findings offer exploratory insights into the synchronization and divergence of European tourism and energy markets, contrasting with the faster recovery dynamics observed in China and the U.S., thus emphasizing the regional disparities in post-pandemic economic resilience and highlighting Europe's slow economic recovery and high energy dependence.

Finally, the wavelet correlation analysis between the LWILDER series and the LAXHRKD series has been performed, and the results are presented in Fig 7.

According to the analysis presented in Fig 7, a strong visual relationship is observed between the two series in the 8–16 month periods (medium term) and 32 + month periods (long term). The figure illustrates that red regions increased after 2020, indicating a notably elevated oscillatory association. Specifically, the medium-term cycles may reflect Australia's growing sensitivity to global energy prices amid economic stability and rising energy demand, particularly during the 2016–2019 period, driven by increased commodity exports and growth in the tourism sector. In contrast, the 2020–2022 period shows heightened coherence in medium frequencies, otentially corresponding to synchronous economic shocks caused by the COVID-19 pandemic, including travel restrictions, sharp declines in energy demand, and reduced consumer confidence.

The rightward phase arrows observed during this period suggest synchronous downturns and positive correlation, reflecting the mid-term economic impacts of the pandemic and energy price fluctuations on Australia's tourism sector. However, in 2023, the low coherence and irregular phase arrows indicate rapid post-pandemic recovery patterns, possibly driven by Australia's domestic demand-oriented economic structure, relatively stable energy prices, and a fast rebound in the tourism sector. Australia's relatively fast recovery and the declining coherence in 2023 may be attributed to its effective domestic pandemic response, early reopening of borders, and proactive tourism stimulus programs such as the "Holiday Here This Year" campaign. Additionally, Australia's high renewable energy penetration and relative independence from

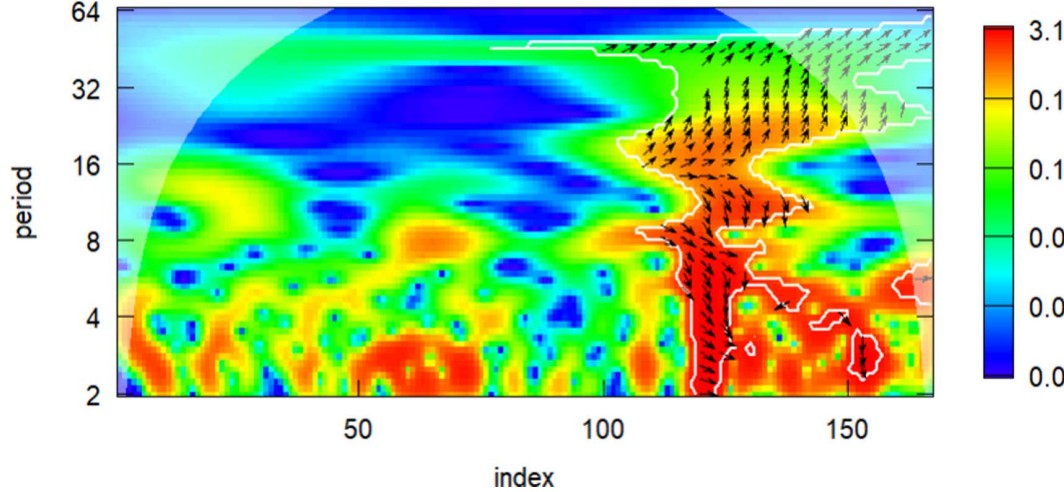

**Fig 7. Wavelet Correlation between the DLWILDER and DLAXHRKD.**

imported fossil fuels may have reduced its exposure to global energy market volatility, allowing for more stable sectoral Dynamics. This may suggest lower sensitivity to global economic uncertainties and a more independent economic recovery trajectory than observed in other regions.

According to the results of the wavelet analysis, periodic bidirectional and complex relationships can be observed between the DWILDER series and the other series between 2010 and 2023. These patterns suggest the presence of mutual co-movement over time. Specifically, the high coherence observed between DWILDER and the STOXX Europe 600 Travel & Leisure Index and the S&P/ASX 300 Hotels Restaurants & Leisure series may imply that the tourism and leisure sectors in Europe and Australia respond similarly to energy price movements, global economic trends, and particularly to the pandemic-related disruptions. The COVID-19 pandemic appears to have contributed to synchronized fluctuations through common channels such as travel restrictions and reduced consumer activity.

These results highlight that shared macroeconomic drivers and investor sentiment may drive comparable adjustment paths in the tourism and clean energy sectors. In particular, the presence of bidirectional patterns at multiple frequency bands underlines the time-varying nature of potential interdependence between clean energy and tourism markets.

## 5. Discussion: interpretation of econometric findings

The results of this study provide empirical evidence on the dynamic and asymmetric relationships between clean energy markets and tourism stock indices across four major economies. While traditional Granger causality tests showed no significant connections, the quantile Granger approach revealed that clean energy shocks affect tourism markets primarily in the upper and lower tails of the return distribution. These findings align with the portfolio diversification theory, which posits that under extreme market conditions such as periods of heightened risk or opportunity investors reallocate capital between sectors. Clean energy stocks, in particular, appear to act as a hedging mechanism during downturns in tourism equity markets, especially in Australia and the U.S.

Moreover, the regional differences highlighted in the wavelet coherence analysis underscore the role of economic structure and policy context in shaping sectoral interactions. For instance, the strong bidirectional relationship between clean energy and tourism in Australia may reflect the country's robust ESG investment environment and ecotourism orientation. In contrast, the weaker and more short-lived coherence observed in the U.S. could be attributed to its domestically driven tourism economy, energy independence, and differentiated monetary policy response during the pandemic. These findings are consistent with previous studies such as [30] and [29], which document regional asymmetries in energy-tourism linkages.

The Chinese tourism market demonstrated relatively faster post-COVID adjustment and stronger medium-run coherence with clean energy movements, which may reflect coordinated stimulus policies and infrastructure-led recovery efforts. In Europe, slower recovery and a more fragmented policy landscape contributed to lagged and less synchronized market dynamics.

From a theoretical standpoint, the observed nonlinearity and time-variation in relationships resonate with the Environmental Kuznets Curve (EKC) framework. As economies progress, clean energy investment and tourism demand may initially conflict, but over time, with institutional development and policy alignment, their interaction can become mutually reinforcing.

Overall, the findings confirm that clean energy and tourism sectors are increasingly interconnected, particularly under systemic shocks such as COVID-19 or major energy price movements. However, the strength, direction, and persistence of these relationships vary significantly across regions, reflecting differences in economic resilience, investor behavior, and sustainability integration.

These insights emphasize the need for tailored policy responses and region-specific portfolio strategies. In line with this, additional evidence from the Structural VAR analysis confirms the existence of regionally differentiated dynamic responses to clean energy shocks. While quantile and wavelet methods capture distribution-sensitive and time-varying

dependencies, the SVAR results provide further clarity on the direction and persistence of structural innovations across different economies.

Specifically, the impulse response functions reveal that China and the United States experience more sustained and positive effects from clean energy shocks, whereas the responses in Europe and Australia appear weaker or indicative of transitional adjustment. These SVAR-based findings reinforce the central argument that clean energy–tourism linkages are both asymmetric and context-dependent, further highlighting the necessity for regionally nuanced policy frameworks. While the inclusion of a Structural VAR model enhances the credibility of causal interpretation, it is important to note that certain econometric limitations such as omitted variable bias or simultaneity may still persist. Future research could benefit from alternative identification strategies including heteroskedasticity-based approaches [48], instrumental variables, or exogenous policy shocks (e.g., renewable energy subsidies) to more rigorously address endogeneity concerns and improve structural inference.

Policymakers may leverage renewable energy incentives not only to achieve environmental goals but also to support tourism sector recovery in times of global volatility. Meanwhile, investors may consider clean energy assets as counter-cyclical hedges against tourism-related risks.

## 6. Conclusion and recommendations

Complementing the quantile and wavelet-based findings, the Structural VAR analysis conducted in the final stage of the study reveals additional empirical support for the existence of dynamic and region-specific transmission mechanisms. By isolating structural shocks and analyzing impulse response behavior, the SVAR results highlight consistent medium-term positive effects in China and the United States, and more muted or complex adjustment paths in Europe and Australia. These patterns validate the broader findings regarding heterogeneity in clean energy-tourism interactions and add robustness to the study's methodological framework.

This study explores the dynamic interrelationship between clean energy and tourism-related stock indices across four major regions using time-series and frequency-domain techniques. By applying Quantile Granger causality and Wavelet Coherence Analysis, the research identifies complex, nonlinear, and time-varying patterns of co-movement that may not be captured by traditional econometric approaches. The findings suggest regionally heterogeneous and quantile-dependent linkages between green energy investments and tourism equities.

The results indicate that clean energy markets exhibit asymmetric co-movement with tourism indices, particularly during periods of global shocks such as the COVID-19 pandemic or major energy price fluctuations. Regional variations may reflect differences in structural economic conditions, policy orientations, and investor sentiment across the United States, Europe, China, and Australia.

## 7. Policy and investment recommendations

Structural impulse responses further suggest that the tourism sector in some regions (e.g., China and the U.S.) may benefit more substantially from sustained clean energy transitions, offering a compelling rationale for aligning environmental policy with economic resilience strategies.

For Governments: Policymakers may consider integrating renewable energy investment incentives with tourism recovery frameworks particularly in environmentally sensitive areas or where tourism constitutes a significant share of GDP. ESG-linked financing instruments and tax incentives could simultaneously advance both climate goals and tourism sector revitalization.

For Asset Managers: Clean energy stocks may serve as hedging instruments against tourism sector volatility during global crises. Incorporating clean energy indices into multi-sector portfolio strategies could enhance diversification and improve risk-adjusted performance, especially in regions like Australia and the U.S. where stronger intersectoral dependencies were observed.

For Tourism Firms: Companies operating in tourism and hospitality could benefit from closely monitoring trends in clean energy pricing and sustainability policies, which often signal shifts in investor sentiment. Integrating ESG compliance into long-term business strategies may also help attract capital from sustainability-focused investment funds.

While this study advances the understanding of inter-sectoral dynamics, it remains limited to four advanced economies and does not include emerging markets such as India, Brazil, or South Africa. Future research could expand the geographic scope and incorporate sector-specific indicators such as carbon emissions, ESG ratings, or firm-level tourism metrics. Furthermore, analyzing structural breaks over longer time horizons may yield additional insights into regime shifts related to policy change, energy transition, or behavioral adjustments.

In conclusion, this study demonstrates how combining time-frequency and quantile-based econometric approaches with structural analysis can generate valuable insights into the evolving co-movement structure of global financial sectors. As markets grow increasingly interconnected, flexible analytical frameworks and region-specific evidence will be essential for informed policy design and investment decision-making.

## Supporting information

**S1 Data. Minimal dataset used in the analysis.**
(XLSX)

## Author contributions

**Conceptualization:** Ozge Demirkale, Naime İrem Duran.

**Methodology:** Ozge Demirkale, Naime İrem Duran.

**Writing – original draft:** Ozge Demirkale, Naime İrem Duran.

**Writing – review & editing:** Ozge Demirkale, Naime İrem Duran.

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
