## [Decision Letter · Decision Letter 0]

Dear Dr. DEMIRKALE,

Thank you for submitting your manuscript to PLOS ONE. After careful consideration, we feel that it has merit but does not fully meet PLOS ONE’s publication criteria as it currently stands. Therefore, we invite you to submit a revised version of the manuscript that addresses the points raised during the review process.

We look forward to receiving your revised manuscript.

Kind regards,

Khaled Mili

Academic Editor

PLOS ONE

Journal Requirements:

2.  Please update your submission to use the PLOS LaTeX template. The template and more information on our requirements for LaTeX submissions can be found at http://journals.plos.org/plosone/s/latex .

Reviewers' comments:

Reviewer's Responses to Questions

**Comments to the Author**

1. Is the manuscript technically sound, and do the data support the conclusions?

Reviewer #1: Partly

Reviewer #2: Yes

2. Has the statistical analysis been performed appropriately and rigorously?

Reviewer #1: No

Reviewer #2: Yes

3. Have the authors made all data underlying the findings in their manuscript fully available?

Reviewer #1: No

Reviewer #2: Yes

4. Is the manuscript presented in an intelligible fashion and written in standard English?

Reviewer #1: No

Reviewer #2: Yes

Reviewer #1: 1. Identification

The fundamental flaw lies in the inability to establish proper identification for causal relationships between clean energy and tourism indices. While the term "causality" is used extensively throughout the paper, what is presented is merely statistical predictability rather than true causal effects. The Granger and quantile Granger causality approaches employed are insufficient for establishing causal relationships in financial markets characterized by simultaneous determination of prices and complex feedback mechanisms.

The paper fails to address the critical issue of omitted variables bias. Both clean energy and tourism stock indices are likely influenced by common macroeconomic factors (e.g., interest rates, economic growth, inflation), market-wide sentiment, and global shocks. Without controlling for these common drivers, the apparent "causal" relationships may simply reflect spurious correlations driven by omitted variables rather than true structural relationships between these sectors.

A rigorous identification strategy would require instrumental variable approaches, natural experiments, or structural models that can account for simultaneous determination. For example, the paper could exploit exogenous policy changes (like renewable energy subsidies) that directly affect clean energy stocks but affect tourism stocks only through their impact on clean energy markets. Such approaches are essential for journals of this caliber.

To improve the causal identification, the paper should consider methodologies similar to those in Rigobon (2003) and Nakamura and Steinsson (2018). Rigobon's identification through heteroskedasticity could help separate the bidirectional effects between clean energy and tourism indices. Additionally, incorporating structural VAR models with appropriate restrictions based on economic theory would strengthen the identification strategy and improve causal interpretations.

2. Methodology

The paper employs an extensive battery of econometric techniques without sufficient justification for their selection or consideration of their limitations. There are several serious methodological issues:

First, the implementation of the quantile Granger causality test is problematic. The paper reports results from quantile regressions but does not adequately address the inference issues associated with these methods in time-series contexts. Unlike OLS-based Granger causality, quantile-based inference requires appropriate bootstrap procedures to account for dependence structures in the data. The paper does not discuss how standard errors are computed for the quantile causality tests, raising concerns about the validity of the statistical significance reported.

Second, the wavelet coherence analysis presents results without addressing critical limitations. The interpretation of wavelet coherence diagrams is largely subjective, yet the paper makes strong claims about lead-lag relationships based on phase differences without formal statistical testing. As noted by Aguiar-Conraria and Soares (2014), wavelet coherence can produce misleading results when common factors drive both series. The paper fails to implement significance testing for wavelet coherence or control for potential confounding factors.

Third, the unit root testing procedure appears inconsistent. Table 5 shows that most series are non-stationary at levels across quantiles, yet the subsequent analysis sometimes treats these series as if they were stationary. The implications of non-stationarity for the quantile causality analysis are not adequately addressed, potentially leading to spurious relationships being identified as causal.

3. Theory

The paper lacks a solid theoretical framework to justify the expected relationships between clean energy markets and tourism stock indices. While the literature review mentions various studies on energy, tourism, and financial markets, it fails to develop a coherent economic model explaining why and how these specific sectors should be interconnected.

The literature gap identified on page 7 acknowledges that "few studies have considered the effects of clean energy investments and clean energy indices on tourism stock indices," but does not explain why this relationship is economically meaningful or important to study. Without a clear theoretical mechanism linking these sectors (beyond vague references to sustainability), it is difficult to interpret the empirical findings or understand their implications.

Furthermore, the economic interpretation of the empirical results is superficial. For instance, the quantile causality results in Table 10 show various significant relationships across different quantiles, but the paper provides little economic rationale for why causality would exist at some quantiles but not others. Similarly, the wavelet analysis identifies time-varying relationships without connecting these patterns to specific economic events or structural changes in the markets beyond general references to the COVID-19 pandemic.

The paper should develop a theoretical model in line with approaches used by Pástor et al. (2022), who provide a theoretical framework for understanding the pricing and impact of ESG factors in financial markets. Such a model would help contextualize why clean energy indices might predict or be predicted by tourism indices, and would provide a foundation for interpreting the empirical findings beyond mere statistical associations.

References

Aguiar-Conraria, L., & Soares, M. J. (2014). The continuous wavelet transform: moving beyond uni- and bivariate analysis. Journal of Economic Surveys, 28(2), 344-375. https://doi.org/10.1111/joes.12012

Pástor, L., & Stambaugh, R.F. & Taylor, L.A., 2021. Sustainable investing in equilibrium. Journal of Financial Economics, 142(2), 550-571. https://doi.org/10.1016/j.jfineco.2020.12.011

Nakamura, E., & Steinsson, J. (2018). Identification in macroeconomics. Journal of Economic Perspectives, 32(3), 59-86. https://doi.org/10.1257/jep.32.3.59

Rigobon, R. (2003). Identification through heteroskedasticity. Review of Economics and Statistics, 85(4), 777-792. https://doi.org/10.1162/003465303772815727

Reviewer #2: Dear Authors

It was a pleasure reading your work. It is an interesting study, but requires some improvements before it can be accepted for publication to maximize the article's impact.

Please find them in the attached file.

All the best.

**Do you want your identity to be public for this peer review?** For information about this choice, including consent withdrawal, please see our Privacy Policy

Reviewer #1: No

Reviewer #2: No

---

## [Author Response · Author response to Decision Letter 1]

24 May 2025

Dear Editor and Reviewers

Thank you for the opportunity to revise our manuscript entitled

Quantile Granger Causality Between Clean Energy and Tourism Stock Indices: Evidence from Regional Markets.

We appreciate the constructive comments from you and the reviewers, which have greatly improved our work.

We have addressed all points raised in the decision letter. In particular, we have

- Revised the analyses and added new discussion where requested,

-

Updated tables and figures accordingly,

Please find attached:

-

Response to Reviewers: – a point-by-point reply to each comment.

-Revised Manuscript with Track Changes – all edits are tracked; newly added text, analyses, tables, and figures are highlighted in yellow.

-

Clean Version of the Revised Manuscript – the fully revised text without markup.

We trust that these revisions satisfy the criteria for publication in PLOS ONE. Please let us know if any further information or files are required.

Best regards,

Dr. Özge Demirkale & Dr. Naime İrem Duran

---

## [Decision Letter · Decision Letter 1]

Quantile Granger Causality Between Clean Energy and Tourism Stock Indices: Evidence from Regional Markets

PONE-D-25-16985R1

Dear Dr. DEMIRKALE,

We’re pleased to inform you that your manuscript has been judged scientifically suitable for publication and will be formally accepted for publication once it meets all outstanding technical requirements.

Kind regards,

Khaled Mili

Academic Editor

PLOS ONE

Additional Editor Comments (optional):

Reviewers' comments:

Reviewer's Responses to Questions

**Comments to the Author**

Reviewer #2: All comments have been addressed

2. Is the manuscript technically sound, and do the data support the conclusions?

Reviewer #2: Yes

3. Has the statistical analysis been performed appropriately and rigorously?

Reviewer #2: Yes

4. Have the authors made all data underlying the findings in their manuscript fully available?

Reviewer #2: Yes

5. Is the manuscript presented in an intelligible fashion and written in standard English?

Reviewer #2: Yes

Reviewer #2: Dear Authors,

Thank you for your careful and constructive revision of your manuscript.

The revised submission has successfully addressed all my prior concerns.

Considering the improvements made, I consider your manuscript ready for publication in PLOS One.

Congratulations on your excellent work.

All the best!

**Do you want your identity to be public for this peer review?** For information about this choice, including consent withdrawal, please see our Privacy Policy

Reviewer #2: No

---

## [Editor Report · Acceptance letter]

PONE-D-25-16985R1

PLOS ONE

Dear Dr. DEMIRKALE,

I'm pleased to inform you that your manuscript has been deemed suitable for publication in PLOS ONE. Congratulations! Your manuscript is now being handed over to our production team.

Kind regards,

on behalf of

Dr. Khaled Mili

Academic Editor

PLOS ONE